**OPINION/HYPOTHESIS**
Ecological and Evolutionary Science
# Emergence of Metabolite Provisioning as a By-Product of Evolved Biological Functions

Sebastian Gude,[a] Gordon J. Pherribo,[a] Michiko E. Taga[a]

[a]Department of Plant and Microbial Biology, University of California, Berkeley, Berkeley, California, USA

Sebastian Gude and Gordon J. Pherribo contributed equally to this work. Sebastian Gude initially suggested writing the manuscript and is listed first.

**ABSTRACT** Microbes commonly use metabolites produced by other organisms to compete effectively with others in their environment. As a result, microbial communities are composed of networks of metabolically interdependent organisms. How these networks evolve and shape population diversity, stability, and community function is a subject of active research. But how did these metabolic interactions develop initially? In particular, how and why are metabolites such as amino acids, cofactors, and nucleobases released for the benefit of others when there apparently is no incentive to do so? Here, we discuss the hypothesis that metabolite provisioning is not itself adaptive but rather can be a natural consequence of other evolved biological functions. We outline two examples of metabolite provisioning as a by-product of other functions by considering cell lysis and regulated metabolite efflux outside their canonical roles and explore their potential to facilitate the emergence of interdependent metabolite sharing.

**KEYWORDS** metabolite provisioning, interdependent metabolism, auxotrophy, leaky functions, metabolite release, intracellular metabolites, lysis, regulated metabolite efflux, microbial evolution, free-living bacteria, microbial communities, coevolution

Existence in the microbial world is contingent on the ability to compete in environments where even the tiniest advantage can mean the difference between survival and extinction. One way to gain an advantage over competitors is to use metabolites produced by others rather than performing all necessary metabolic functions independently. Indeed, most microbes depend on others for some metabolites and, at the same time, release metabolites into the environment (1, 2). These interactions collectively form complex networks of interconnected metabolic pathways among numerous species (Fig. 1A). Of particular curiosity is the origin of these interactions: how does metabolic interdependence arise in a competitive world?

Dependence on others to fulfill core metabolic requirements starts with the loss of metabolic capabilities via loss-of-function mutations (Fig. 1B and C). The emergence and maintenance of loss-of-function mutants can be explained by the Black Queen Hypothesis, which argues that the loss of a gene is favored by natural selection as long as the encoded function is sufficiently "leaky" and, hence, continuously fulfilled by others (3). Functions that change the extracellular environment, such as degradation of toxins, polysaccharide hydrolysis, and release of iron-scavenging siderophores, are inherently leaky; therefore, organisms that do not perform these functions can still reap their benefits (Fig. 2A) (4–6). Another type of leakiness is the excretion of temporarily undesired, yet energetically valuable, metabolic by-products such as acetate and lactate (overflow metabolism) (Fig. 2B) (7, 8). Hence, secondary effects of biological processes, which evolved to benefit the organism performing it, can also provide benefits to unrelated organisms.

Unlike the examples above, metabolic end products such as amino acids, cofactors,

Address correspondence to Michiko E. Taga, taga@berkeley.edu.

Nearly all microbes benefit from using metabolites produced by others. But how do microbes evolve to provide nutrients to others in their community? Metabolite provisioning can evolve as a byproduct of other evolved functions.

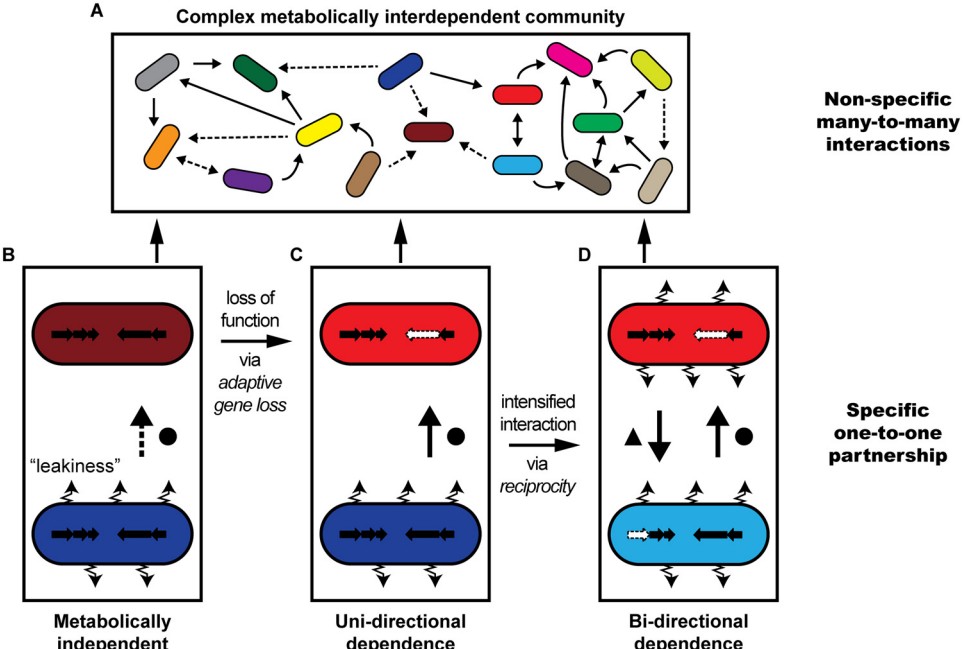

**FIG 1** Structure and evolution of metabolically interconnected microbial communities. (A) Microbes form complex networks of metabolic interactions in which many metabolites are shared among many species. (B to D) Evolution of specific one-to-one partnerships. (B) Generalists (dark red and dark blue) synthesize their own metabolites and thus are metabolically independent but can engage in facultative interactions. (C) A loss-of-function mutation via adaptive gene loss in one population (light red, auxotroph) is compensated by an obligate interaction with a "leaky" producer population (dark blue). (D) Reciprocity facilitates intensified cooperation, resulting in a bidirectional obligate interaction. The original auxotroph (light red) supplies a growth-promoting metabolite to the original producer (light blue) which has evolved dependence on its partner. Facultative interactions are represented by dashed arrows and obligate interactions by solid arrows in all panels.

and nucleobases are not considered inherently leaky because they are produced and used inside the cell. However, genomic analyses indicate that the majority of microbes are unable to produce one or more amino acids, cofactors, or nucleobases (i.e., are auxotrophs) (9–11), suggesting that these intracellular metabolites are somehow leaky or that dedicated mechanisms of releasing them are prevalent. Auxotrophy and extensive genome reduction commonly occur in obligate endosymbionts, driven by genetic drift caused by genetic isolation and the bottlenecks that occur during transmission. Auxotrophy also commonly occurs in free-living microbes, though to a lesser extent, suggesting that auxotrophy can also arise via natural selection. In these relatively rare cases, auxotrophy may be supported by leakiness of other organisms in the environment (10).

In laboratory experiments, growth of some amino acid auxotrophs of *Escherichia coli* is supported by a partner in coculture, suggesting that actively proliferating microbial populations can be leaky for some intracellular metabolites (9). This suggests that nutrients can be provided within free-living microbial communities in the absence of a host. The mechanisms by which these metabolites are released in free-living microbial populations remain largely unexplored. How and why are these important and energetically valuable products released from cells? How can a seemingly non-beneficial behavior arise in free-living microbes? In other words, "what's in it for the producer?"

Provisioning of resources to others can be beneficial to the producing population in reciprocal interactions: when one population provides a growth-enhancing metabolite to another, it may be rewarded with a larger amount of its required metabolite in return (Fig. 1C and D) (12). Thus, once a single metabolic dependence has arisen, the subsequent evolutionary steps toward additional metabolic interdependences are thought to be adaptive (13). Furthermore, for these interactions to be stable, effective mechanisms to exclude cheaters (individuals that benefit without contributing) must

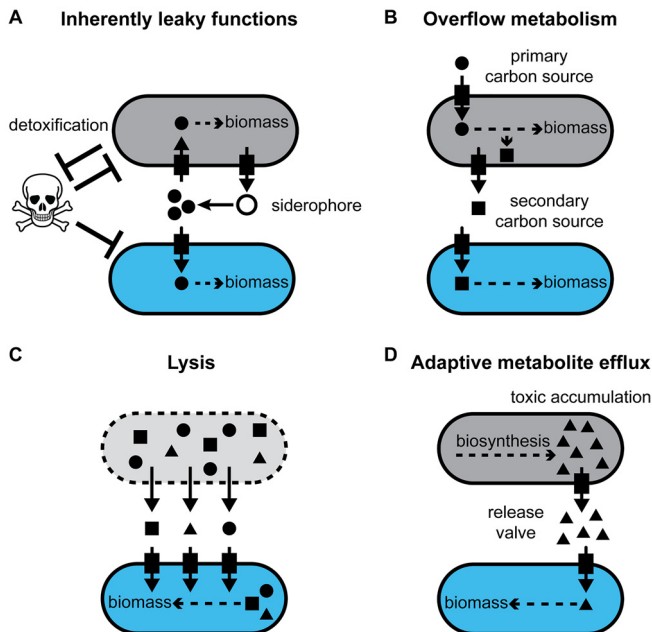

**FIG 2** Mechanisms of metabolite provisioning via "leakiness." (A) Some cellular functions are inherently leaky since they are performed outside the cell. Examples include enzymatic detoxification of the environment, such as antibiotic degradation, and nutrient acquisition, such as siderophore secretion. (B) Secondary carbon sources are released during overflow metabolism to maximize growth rate in the producer population (gray). (C) Lysis, resulting from viral predation or other natural processes, releases intracellular contents into the environment. (D) Accumulation of intracellular metabolites, which can inhibit growth, can be alleviated by specific metabolite efflux systems that function as a "release valve."

be present. Yet still unclear is how leakiness for intracellular metabolites is initially established to allow for the emergence of metabolic dependence. For this initial evolutionary step to occur in free-living microbial populations, a route for metabolite release must be present. In addition, two criteria must be fulfilled. First, the mechanism of overproducing these desirable metabolites must not incur a fitness cost to the producer, since there is no reciprocation to compensate for it. Second, a sustained supply of the metabolite must be maintained in order for auxotrophs to become fixed in the population. Here, we outline how by-products of evolved biological functions can lead to a sustained provisioning of intracellular metabolites, including energetically valuable amino acids, in free-living non-host-associated microbial communities. We employ the term "by-product" to mean any secondary effect of a biological process (14) rather than limiting its use to metabolic by-products (13). In the following sections, we discuss two examples of metabolite provisioning mechanisms by considering lysis and regulated metabolite efflux outside their canonical roles (Fig. 2C and D).

## SUSTAINED METABOLITE PROVISIONING AS A BY-PRODUCT OF LYSIS

Lysis, a death process resulting from a loss of cell envelope integrity, is a fundamental part of microbial life. Microbial cells lyse when experiencing stress and can be killed by competing microbes (15, 16). Microbes also undergo lysis as part of developmental programs such as biofilm formation, fruiting-body development, and sporulation (17–19). One of the most common mechanisms of lysis across diverse environments is mediated by viruses (20, 21). Viral lysis is a key factor in modulating natural microbial ecosystems. In marine environments, viruses lyse an estimated 20% to 40% of microbial biomass each day, making them major drivers of phytoplankton mortality (21). The importance of viruses in modulating microbial communities has also been demonstrated in controlled laboratory systems where viral predation was shown to alter competitive interactions between microbial species (22).

Any lysis process inherently results in the release of intracellular material. Therefore,

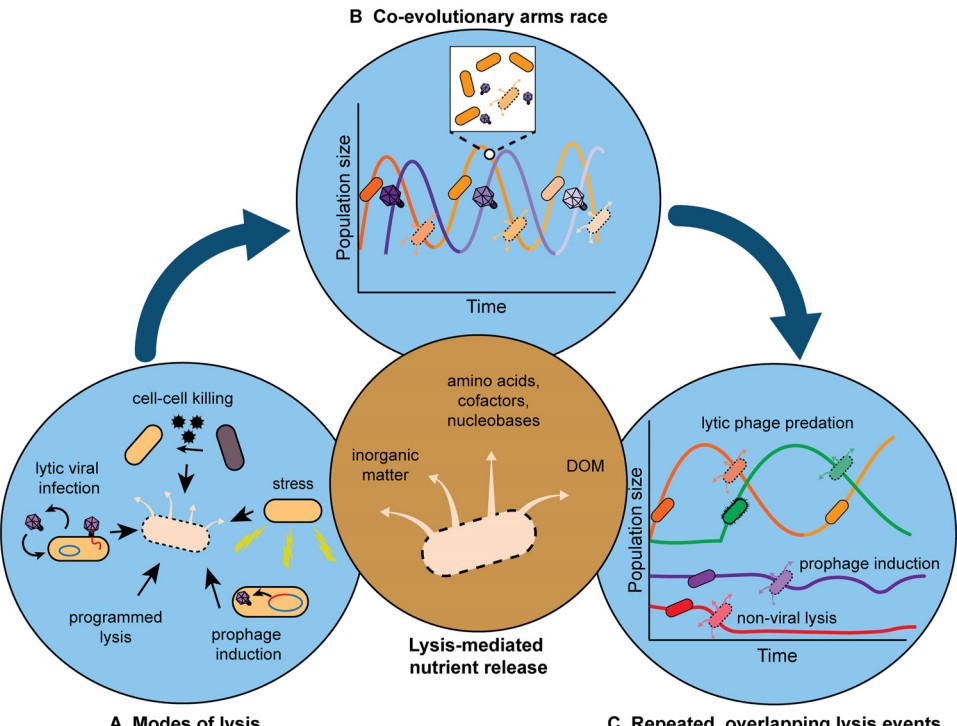

**FIG 3** Lysis as a mechanism of nutrient provisioning. Lysis (center) releases inorganic matter (e.g., iron, phospho-rus, and selenium), dissolved organic matter (DOM), and other intracellular metabolites synthesized in the cell (e.g., amino acids, cofactors, and nucleobases) into the environment. (A) Different modes of lysis contribute to nutrient release. (B) Viruses and their hosts coevolve via Red Queen dynamics, resulting in population size and nutrient release fluctuations (boom-and-bust cycles). (C) Overlapping lysis events across different populations can provide a sustained supply of metabolites.

lysis may be considered a mechanism of "leakiness" for metabolites that are normally contained within cells (Fig. 2C). The bioavailability of these cellular components may vary depending on the character of the molecule and how an organism accesses the nutrient. For example, the amino acids or metabolites sequestered within proteins can only be made available if proteases are present. Evidence for the release of bioavailable nutrients following cell lysis has been observed across various scales and systems. Release of iron, nitrogen, and carbon resulting from lysis has been shown to support microbial growth in aquatic environments, demonstrating that these cellular compo-nents are present in a bioavailable form (Fig. 3A) (23, 24). Additional support for the availability of nutrients following lysis has been established in laboratory experiments. T7 phage lysis of *E. coli* was shown to support higher growth yields of *Salmonella enterica* serovar Typhimurium in coculture than under conditions without phage, indicating the release of bioavailable nutrients, including a carbon source, via lysis (25). Similarly, when nutrient deprived, a subpopulation of *Bacillus subtilis* cells undergoes programmed lysis to provide nutrients to their kin in order to delay sporulation (18). Together, these findings demonstrate that nutrients liberated via lysis can support the growth of microbes within and across species and suggest that these nutrients can also contribute to the emergence of auxotrophs.

An individual lysis event results only in a transient release of nutrients, but for the emergence and maintenance of auxotrophic populations, a sustained nutrient supply is needed. One framework in which lysis can generate a sustained nutrient supply is the coevolutionary arms race between viruses and their hosts. Red Queen dynamics predict the emergence of virus-resistant microbial hosts following a lysis-induced collapse of the host population (26). This recovery lasts only until the host cells are again lysed by a newly evolved viral mutant, thus leading to repeated lysis events (Fig. 3B). Laboratory evolution studies indeed have observed such sustained coevolutionary dynamics in

phage-bacteria systems (27). These boom-and-bust cycles created by an individual virus-host pair will thus produce a fluctuating supply of nutrients as a by-product of lysis. However, virus-host coevolution is not limited to a single virus-host pair but occurs in the larger context of complex communities in which numerous viruses and hosts naturally exist. According to the kill-the-winner hypothesis, the most abundant microbial population within a community will be preyed upon most frequently, causing its population to collapse (28). After this population collapse, a different microbial population will become most abundant and likewise experience a higher susceptibility to viral predation and lysis. Population dynamics like these, coupled to coevolutionary arms race dynamics in the context of large microbial communities, are thus capable of creating overlapping and alternating boom-and-bust cycles (Fig. 3C). We propose that many uncoordinated repeated lysis events can together provide a sustained supply of nutrients to enable the emergence and maintenance of auxotrophs. To date, limited experimental evidence for coupled oscillations within virus-microbe systems exists; *in situ* measurements of microbial and viral abundances with improved time scale resolution could provide useful insight into these dynamics (29).

Coevolutionary arms race dynamics between viruses and their microbial hosts may halt for any particular virus-host pair, such as when microbes evolve full resistance (30) or when environmental factors modulate species interactions (31). However, many other mechanisms of cell lysis exist and can also contribute to nutrient provisioning. For example, bacterial lysis is occasionally activated from within the host genome via prophage induction (32), and lysis is also caused by non-viral mechanisms, such as autolysis induced by stress or as part of a developmental process and killing due to microbial competition (Fig. 3A and C) (15–19). Autolysis has been shown to be important in supporting biofilm development by releasing DNA to form a structural component of extracellular matrices; this process could also support neighboring cells through released nutrients (19). Thus, various combinations of these processes within large and diverse microbial communities can produce a sustained nutrient supply by creating many interspersed nutrient release events (Fig. 3A and C). Consequently, we posit that intracellular metabolites such as amino acids, cofactors, and nucleobases can perpetually be made available to support the emergence and maintenance of auxotrophs in microbial communities as a by-product of evolved mechanisms of lysis.

## METABOLITE EFFLUX AS A BY-PRODUCT OF MAINTAINING HOMEOSTASIS

Efflux systems in microbes have long been appreciated as mechanisms to protect cells from the accumulation of toxins and metabolic waste products, excrete small molecules such as siderophores and flavins to fulfill a specific metabolic requirement, and export building blocks for the assembly of extracellular structures (7, 33–36). Recently, microbes have also been found to possess efflux systems for the excretion of amino acids and other intracellular metabolites (37). These efflux systems have been speculated to act as "release valves" that modulate intracellular metabolite levels to restore homeostasis following temporary metabolic imbalances (Fig. 2D) (37). For example, cysteine exporters in *E. coli* are proposed to protect cells from toxic intracellular accumulation of cysteine, and a homoserine exporter has been suggested to play a role in maintaining homeostatic levels of threonine (37). These processes thus perform a function akin to the release of temporarily undesirable metabolites during overflow metabolism (Fig. 2B), as both protect the cell from growth inhibition due to metabolite accumulation (7, 8).

We speculate that metabolite efflux systems can also provide benefits during unperturbed growth by releasing products accumulated as a result of continuous overproduction (Fig. 2D). Continuous overproduction, though seemingly in contrast to the principle of cellular economy, may be a consequence of the highly interconnected nature of cellular metabolism as well as an adaptive strategy to avoid the negative consequences of underproduction. When even one required metabolite becomes limiting, growth is reduced (Fig. 4, $T_{under}$). However, the highly interconnected nature of metabolic networks fundamentally limits the extent to which metabolite production

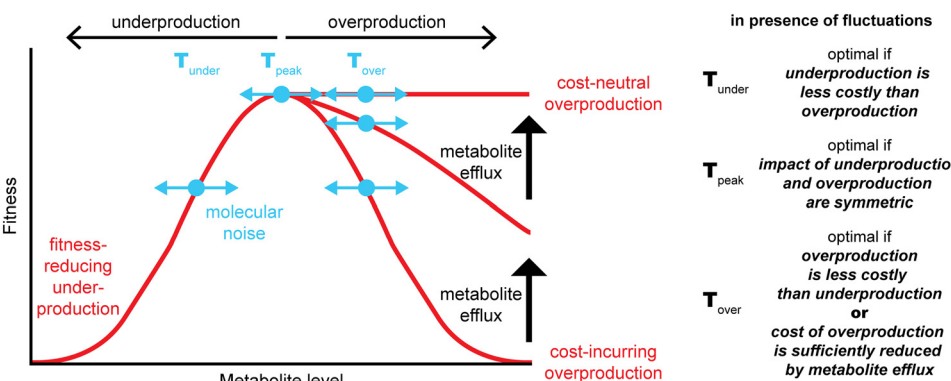

**FIG 4** Regulated metabolite efflux can turn overproduction into an adaptive strategy. Fitness (red lines) is shown as a function of intracellular metabolite level. Underproduction or overproduction of a metabolite results in decreased fitness. Setting production targets (T, blue dots) is limited by inherent fluctuations (blue arrows) and the highly interconnected nature of metabolism. Regulated metabolite efflux can diminish the cost of overproduction by alleviating toxic effects of intracellular metabolite accumulation, shifting the fitness curve toward cost-neutral overproduction (vertical black arrows). Optimality of production targets ($T_{under}$, $T_{peak}$, or $T_{over}$) depends on the exact dependence of fitness on the intracellular metabolite level and the magnitude of fluctuations.

levels can be optimized (38). For example, aspartate is a precursor in alanine and asparagine biosynthesis, and serine is part of glycine, cysteine, and methionine synthesis pathways. Consequently, the stoichiometry of certain sets of metabolites is hard wired by the topology of the metabolic network. To satisfy the minimal requirements for all metabolites at any time, some metabolites need to be produced in excess (38). One strategy for relieving metabolite accumulation is degradation. However, some organisms such as *E. coli* lack degradation pathways for certain metabolites and therefore must remove excess metabolites by exporting them (38). Continuous active export of these overproduced metabolites via designated efflux systems can avert their intracellular accumulation and partially diminish the negative fitness consequences associated with metabolite overproduction (Fig. 4, $T_{over}$, black arrows) (39).

Microbes may also have evolved to overproduce metabolites to buffer against inherent fluctuations in gene expression, enzyme activity, and regulatory systems (Fig. 4, blue arrows). One source of these fluctuations is the variation in gene copy number that occurs during genome replication, resulting in deterministic variations in gene expression levels (40). Another source of fluctuations is the limited capacity of the cytoplasm, where proteins produced at low levels are particularly affected by stochastic effects (41). For example, the average protein abundance in *Mycoplasma pneumoniae* is estimated to be 167 molecules per cell (42), and at least 10% of all proteins in *E. coli* are present at fewer than 10 copies per cell (43). Therefore, if an organism would aim to produce the optimal level of all metabolites required for its growth (Fig. 4, $T_{peak}$), inherent fluctuations in protein levels (Fig. 4, blue arrows) would cause it to sometimes produce too much and at other times too little. Underproduction is always detrimental because it limits growth (Fig. 4, $T_{under}$). Overproduction can also decrease fitness when resources such as precursors and energy are wasted or when metabolites accumulate to toxic levels (Fig. 4, $T_{over}$) (39). The latter effect can be alleviated by actively exporting excess metabolites via designated efflux systems (Fig. 4, black arrows) (37).

Though seemingly wasteful, a strategy in which cells constantly overproduce to fulfill minimal production requirements, and export excess metabolites to avoid accumulation and toxicity, can be adaptive if overproduction is less harmful than underproduction (Fig. 4). Therefore, continuous export of some intracellular metabolites may be a fundamentally adaptive strategy that, as a by-product, fortuitously provides nutrients to auxotrophs. Laboratory experiments have provided hints that some intracellular metabolites may indeed be exported. The growth of lysine, methionine, and phenylalanine auxotrophs of *E. coli* is supported by other *E. coli* mutants in coculture,

reinforcing the idea that some amino acids may be actively exported (9). Furthermore, methionine and phenylalanine auxotrophs fared slightly better than other amino acid auxotrophs in pooled transposon sequencing (Tn-Seq) experiments, again suggesting that these amino acids can be made available by other cells (44).

## CONCLUSION

The pervasiveness of auxotrophy throughout microbial genomes suggests that the release of intracellular metabolites is prevalent. Metabolite provisioning for the benefit of others can arise through various adaptive and non-adaptive mechanisms. For example, metabolites can be provided in the context of coevolved partnerships, where evolved reciprocity or genome reduction leads to metabolic interdependence (Fig. 1C and D). Here, rather than addressing partnerships that have coevolved, such as endo-symbionts and their hosts, we discuss how nutrient-sharing relationships can be initiated in free-living microbes (Fig. 1A). Lysis and metabolite efflux systems are, in principle, capable of creating a sustained supply of metabolites that can support the emergence and maintenance of auxotrophs and therefore may contribute to the evolution of metabolic interdependence. Metabolite provisioning as by-products of evolved functions may present a complementary alternative to our current perception that specific interactions predominantly evolve between specialized partners (Fig. 1B to D) by shifting the focus to non-specific metabolite provisioning that can benefit any organism (Fig. 1A). Thus, metabolite provisioning as a by-product (Fig. 2) can lead to non-specific many-to-many interactions in which multiple organisms provide and take up nutrients, as opposed to specific one-to-one coevolved partnerships. Such many-to-many sharing interactions may present a solution to the paradox of how the initial steps of nutrient sharing can evolve in free-living microbial communities in the absence of partner-specific reciprocal interactions.

## ACKNOWLEDGMENTS

This work was funded by National Institutes of Health grant DP2AI117984 to M.E.T. and by the National Science Foundation Graduate Research Fellowship Grant DGE 1752814 to G.J.P.

We thank Adam Deutschbauer and members of the Taga lab for helpful discussions. We also thank Kristopher Kennedy, Olga Sokolovskaya, Amanda Shelton, Kenny Mok, and Zachary Hallberg for critical reading of the manuscript.

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
