## [Reviewer comments · mSystems]

Emergence of metabolite provisioning as a byproduct of evolved biological functions

Sebastian Gude, Gordon Pherribo, and Michiko Taga

Corresponding Author(s): Michiko Taga, University of California, Berkeley

Review Timeline:

Submission Date:	March 23, 2020
Editorial Decision:	April 17, 2020
Revision Received:	May 11, 2020
Accepted:	May 14, 2020

Editor: Benjamin Wolfe

Reviewer(s): The reviewers have opted to remain anonymous.

Transaction Report:

DOI: <https://doi.org/10.1128/mSystems.00259-20>

April 17, 2020

Dr. Michiko E Taga
University of California, Berkeley
Plant & Microbial Biology
111 Koshland Hall
Berkeley, CA 94720-3102

Re: mSystems00259-20 (Emergence of metabolite provisioning as a byproduct of evolved biological functions)

Dear Dr. Michiko E Taga:

Thank you for submitting your Opinion/Hypothesis piece to mSystems. Two reviewers have provided valuable feedback on your manuscript. They both agreed that the perspective piece is a valuable contribution for the field of microbiology and microbial ecology. They provided important suggestions to improve the clarity and impact of the piece that you should address as you revise your manuscript.

Please also note that the number of references permitted for Opinion/Hypothesis pieces at mSystems is 25. You currently have 64. As you make your revisions, please try to reduce the number of references.

Below you will find the comments of the reviewers.

To submit your modified manuscript, log onto the eJP submission site at <https://msystems.msubmit.net/cgi-bin/main.plex>. If you cannot remember your password, click the "Can't remember your password?" link and follow the instructions on the screen. Go to Author Tasks and click the appropriate manuscript title to begin the resubmission process. The information that you entered when you first submitted the paper will be displayed. Please update the information as necessary. Provide (1) point-by-point responses to the issues raised by the reviewers as file type "Response to Reviewers," not in your cover letter, and (2) a PDF file that indicates the changes from the original submission (by highlighting or underlining the changes) as file type "Marked Up Manuscript - For Review Only."

Due to the SARS-CoV-2 pandemic, our typical 60 day deadline for revisions will not be applied. I hope that you will be able to submit a revised manuscript soon, but want to reassure you that the journal will be flexible in terms of timing, particularly if experimental revisions are needed. When you are ready to resubmit, please know that our staff and Editors are working remotely and handling submissions without delay. If you do not wish to modify the manuscript and prefer to submit it to another journal, please notify me of your decision immediately so that the manuscript may be formally withdrawn from consideration by mSystems.

To avoid unnecessary delay in publication should your modified manuscript be accepted, it is important that all elements you upload meet the technical requirements for production. I strongly recommend that you check your digital images using the Rapid Inspector tool at <http://rapidinspector.cadmus.com/RapidInspector/zmw/>.

Sincerely,

Benjamin Wolfe

Editor, mSystems

Journals Department
Reviewer comments:

Reviewer #1 (Comments for the Author):

Gude, et al. discuss the means by which metabolic interdependence arises via natural selection in free-living bacteria. This review examines the observation that many bacteria are auxotrophs for one or several essential nutrients suggesting that a ubiquitous and constant supply of these nutrients are present in a microbe's environment. Gude, et al. suggests several potential mechanisms by which nutrients are made available as a common good. First, the authors consider cell lysis as a source of nutrients for auxotrophs. Viral predation, a common cause of lysis in the microbial world, is discussed as a means to provide nutrients via necrophagy. The 'kill the winner' hypothesis, postulates that within complex communities, numerically dominant members are preyed upon by viruses more frequently which reduces their population size allowing other minority members to bloom. Boom and bust cycles allow for a constant release of nutrients into the ecosystem. The authors briefly mention other mechanisms of cellular lysis. The next section proposes several scenarios where the efflux of metabolites is incentivized. Metabolite efflux has been suggested to act as a 'release valve' to prevent intracellular accumulation of overproduced metabolites. The authors argue that overproduction of certain key metabolites is less detrimental to a cell's fitness than underproduction because some metabolites are required for other metabolic pathways. In other words, underproduction of certain metabolites would create a metabolic bottleneck. Related to this, stress caused by overproduction can be reduced by efflux. Additionally, because gene expression/protein production can fluctuate and be affected by random events, overproduction of some proteins is selective as underproduction can have larger negative fitness

effects. Together, this review/opinion piece provides several lines of reasoning to explain how interbacterial cross-feeding may arise and why auxotrophy is prevalent among bacteria.

The document is well written. The paper's figures are simple and informative.

Minor suggestions:

1. An additional example of lysis mediated nutrient release is through the production of bacteriocins. Within *E. coli* some bacteriocins (colicins belonging to E and K, for instance) are released by cell lysis. The molecules, in turn, may cause cell lysis in sensitive cells. Like phage, bacteriocins are often highly specific in who they kill. Although the induction of bacteriocins is tightly regulated and highly specific, production of these molecules can be prompted by starvation. Anywhere from 10-70% of *E. coli* isolates depending on the environment carry bacteriocin genes. It seems reasonable to assume that this is true for other bacterial species as well. Bacteriocins would be an additional area to cover with respect to cell lysis in the provisioning of nutrients.

a. A useful resource: Gordon DM. The Natural History of Bacteriocins. The Bacteriocins: Current Knowledge and Future Prospects; Dorit, RL, Roy, SM, Riley, MA, Eds. 2016 Jul 15:1-0.

2. Biofilm formation and programmed cell death is briefly mentioned in the text. Given that most environmental microbes are part of complex polymicrobial biofilms, cell death within these structures may provide a stable source of nutrients for other members of the community. There is at least some evidence that DNA released by cell lysis within biofilms is an integral component of the biofilm matrix. More emphasis on work associated with programmed cell death and biofilms would improve this work as lytic phage abundance and the rate of infection is highly dependent on the environment and likely plays a smaller role than programmed cell death in nutrient provisioning.

a. PMID: 15184116

b. PMID: 17694072

3. Lines 156-168 talk about some metabolites being essential for other pathways in a metabolic network. Please provide a few examples of these metabolites and whether there is any data to suggest that they are present at increased levels in extracellular environments. Based on metabolic reconstructions is it possible to predict what nutrients will be made available? Lines 189-191 mention that lysine, methionine, and phenylalanine auxotrophs are able to survive in co-culture with other *E. coli* suggesting that they may be excreted/released - is there evidence to suggest that these amino acids are 'key metabolites' essential for other pathways and thus are overproduced? If so, elaboration in the text would be appreciated.

4. Are there any metabolomic datasets that back up the claim that nutrients are present at levels high enough to support auxotrophs? If so, it may be worth mentioning.

Reviewer #2 (Comments for the Author):

mSystems00259-20 - Emergence of metabolite provisioning as a byproduct of evolved biological functions

The opinion/hypothesis "Emergence of metabolite provisioning as a byproduct of evolved biological functions" by Gude, Pherribo, and Taga describes the concept that metabolite provisioning is not an evolutionary adaptive phenomenon but rather a natural biological consequence. The authors

describe two examples of metabolic provisioning mechanisms as the main mechanisms that allowed the emergence of inter-dependencies. The manuscript is well written and easy to follow. However, the idea of leakiness by itself allowing the emergence of metabolic dependencies, seems very unlikely. The authors should point out the contradictions and disadvantages of adaptive mechanisms, contra pointing both hypotheses. Why they could not happen concomitantly?

Major comments:

Lines 20 and 51: The generic use of "nucleic acid auxotrophy" can be misleading. To my knowledge all organisms are autotrophic for DNA and RNA. Please review the text and provide more explanation, references.

Lines 61-62: The authors highlight the absence of knowledge of the mechanisms used by microbes to share nutrients. To date, it is known that most nutrients are shared through direct uptake from the outer environment. The authors cited a manuscript describing formation of nanotubes between microbes. It seems an odd choice since it is not known to be a common mechanism for exchanging intra-species.

Line 124: "Host population" usage is confusing here. Make clear you are talking about phages bacterial hosts. In addition, what would it be, bacterial species, strains?

Line 124-131: How kill-the-winner hypothesis supports microbiota homeostasis? We have several examples of how stable microbial communities can be over time, i.e, the most abundant members stay the same unless perturbations happen. The idea of many uncoordinated lysis events occurring is intriguing, could the author suggest an approach to validate it?

Line 138: "various viral and non-viral lysis processes", the authors should briefly cite them and explain why they are less relevant than the viral lysis process.

Lines 145, 67-68: The concept of the overproduction of metabolites has to be better explained. In regulatory processes, small changes in metabolites concentrations can dramatically affect the organism response and byproducts can impair growth (PMID: 30988035, 18618908) Additionally, how microbes would handle growth inhibition in the first place? Same question came when I was reading the two criteria for the occurrence of the initial evolutionary step (Line 73). Overproductions of metabolites and sustained supply of nutrients without a efflux route would promote growth inhibition.

Minor comments:

Line 43: "biopolymer degradation" as a naturally occurring process outside the cell. It seems unusual for me and the references listed do not support it.

Line 80: The use of the term "side effect" for byproducts is not adequate, it gives an impression of harm. Maybe replace by "final result" or "output".

Line 89: No references for biofilm formation, all sporulation.

Line 222: Check references format. No capital letter and the bacterial names should be italicized.

Green - Responses

Reviewer #1 (Comments for the Author):

Gude, et al. discuss the means by which metabolic interdependence arises via natural selection in free-living bacteria. This review examines the observation that many bacteria are auxotrophs for one or several essential nutrients suggesting that a ubiquitous and constant supply of these nutrients are present in a microbe's environment. Gude, et al. suggests several potential mechanisms by which nutrients are made available as a common good. First, the authors consider cell lysis as a source of nutrients for auxotrophs. Viral predation, a common cause of lysis in the microbial world, is discussed as a means to provide nutrients via necrophagy. The 'kill the winner' hypothesis, postulates that within complex communities, numerically dominant members are preyed upon by viruses more frequently which reduces their population size allowing other minority members to bloom. Boom and bust cycles allow for a constant release of nutrients into the ecosystem. The authors briefly mention other mechanisms of cellular lysis. The next section proposes several scenarios where the efflux of metabolites is incentivized. Metabolite efflux has been suggested to act as a 'release valve' to prevent intracellular accumulation of overproduced metabolites. The authors argue that overproduction of certain key metabolites is less detrimental to a cell's fitness than underproduction because some metabolites are required for other metabolic pathways. In other words, underproduction of certain metabolites would create a metabolic bottleneck. Related to this, stress caused by overproduction can be reduced by efflux. Additionally, because gene expression/protein production can fluctuate and be affected by random events, overproduction of some proteins is selective as underproduction can have larger negative fitness effects. Together, this review/opinion piece provides several lines of reasoning to explain how interbacterial cross-feeding may arise and why auxotrophy is prevalent among bacteria.

The document is well written. The paper's figures are simple and informative.

Minor suggestions:

1. An additional example of lysis mediated nutrient release is through the production of bacteriocins. Within *E. coli* some bacteriocins (colicins belonging to E and K, for instance) are released by cell lysis. The molecules, in turn, may cause cell lysis in sensitive cells. Like phage, bacteriocins are often highly specific in who they kill. Although the induction of bacteriocins is tightly regulated and highly specific, production of these molecules can be prompted by starvation. Anywhere from 10-70% of *E. coli* isolates depending on the environment carry bacteriocin genes. It seems reasonable to assume that this is true for other bacterial species as well. Bacteriocins would be an additional area to cover with respect to cell lysis in the provisioning of nutrients.

- a. A useful resource: Gordon DM. The Natural History of Bacteriocins. The Bacteriocins: Current Knowledge and Future Prospects; Dorit, RL, Roy, SM, Riley, MA, Eds. 2016 Jul 15:1-0.

Response: We agree that microbial competition via bacteriocins is indeed an interesting and important mode of bacterial lysis that would be worth describing in detail. We considered adding more content on bacteriocins, but ultimately did not do so, in part because we were asked to limit the number of references. With this in mind, we do indirectly refer to processes like bacteriocins by stating that microbes “can be killed by competing microbes” (line 88-89). One of the references cited (Ghoul et al, 2016) discusses bacteriocins in the context of microbial competition.

2. Biofilm formation and programmed cell death is briefly mentioned in the text. Given that most environmental microbes are part of complex polymicrobial biofilms, cell death within these structures may provide a stable source of nutrients for other members of the community. There is at least some evidence that DNA released by cell lysis within biofilms is an integral component of the biofilm matrix. More emphasis on work associated with programmed cell death and biofilms would improve this work as lytic phage abundance and the rate of infection is highly dependent on the environment and likely plays a smaller role than programmed cell death in nutrient provisioning.

a. PMID: 15184116*

b. PMID: 17694072

Response: Thank you for pointing us to these references. We now cite PMID: 15184116 (line 90, 142 and 145). We have also added the following statement to emphasize the potential contribution of autolysis in biofilm formation as a non-viral mechanism of nutrient release:

Line 142-145: ‘Autolysis has been shown to be important in supporting biofilm development by releasing DNA to form a structural component of biofilm extracellular matrices; this process could also support neighboring cells through released nutrients (19).’

3. Lines 156-168 talk about some metabolites being essential for other pathways in a metabolic network. Please provide a few examples of these metabolites and whether there is any data to suggest that they are present at increased levels in extracellular environments. Based on metabolic reconstructions is it possible to predict what nutrients will be made available? Lines 189-191 mention that lysine, methionine, and phenylalanine auxotrophs are able to survive in co-culture with other E. coli suggesting that they may be excreted/released - is there evidence to suggest that these amino acids are 'key metabolites' essential for other pathways and thus are overproduced? If so, elaboration in the text would be appreciated.

Response: We now mention two specific examples in amino acid biosynthesis, highlighting the high degree of interconnectedness in some metabolic pathways:

Line 172-174: 'For example, aspartate is a precursor in alanine and asparagine biosynthesis, and serine is part of glycine, cysteine, and methionine synthesis pathways.'

To our knowledge there is limited experimental data to suggest that amino acids, cofactors, and nucleobases could be present at increased levels in natural extracellular environments. Paczi et al. 2012 (<https://doi.org/10.1186/1475-2859-11-122>) did detect amino acids and intermediates from central carbon metabolism in batch culture media. Hypotheses for the presence of these metabolites in the extracellular environment range from protein digestion, destruction of cell wall integrity during sampling to overflow metabolism. Yet it remains unclear how these findings relate to natural habitats (see also point 4).

There is no evidence that lysine, methionine, and phenylalanine are 'key metabolites' (as defined by the reviewer). For example, lysine is not a precursor for any other amino acid. We believe the key factor determining which metabolites are overproduced and accumulate is likely an emergent property of the global flux distribution within a metabolic network.

In recent years computational studies have started to explore the potential for metabolite sharing in microbial communities with the help of metabolic reconstructions. (<https://doi.org/10.1371/journal.pcbi.1003695>, <https://doi.org/10.1186/s12918-018-0588-4>, <https://doi.org/10.1038/s41467-018-07946-9>). Interestingly, one study predicts cost-free production of nucleic acids and peptides (<https://doi.org/10.1038/s41467-018-07946-9>). Yet, another study (<https://doi.org/10.1186/s12918-018-0588-4>) specifically mentions the poor agreement of their predictions with experimental observations indicating the limited predictive power of current metabolic models with respect to the specifics of metabolite sharing. Consequently, we did not mention metabolic reconstructions in the text. We hope that future iterations of metabolic reconstructions will be more predictive. A more detailed discussion of some of these metabolic modeling studies can be found in our recent review (<https://doi.org/10.1016/j.copbio.2019.08.005>).

4. Are there any metabolomic datasets that back up the claim that nutrients are present at levels high enough to support auxotrophs? If so, it may be worth mentioning.

Response: We are not aware of general metabolomic characterizations of environments inhabited by free-living microbes. We agree that such information would be very valuable to add.

Focused studies in aquatic systems indicate environmental vitamin B12 (a cofactor), and the presence of vitamin B12 auxotrophs in this environment suggests that vitamin B12 levels are sufficient to support growth (<https://doi.org/10.1073/pnas.1608462114>). The sharing of vitamin B12 in aquatic environments has previously been discussed elsewhere (<https://doi.org/10.1128/EC.00097-06>, <https://doi.org/10.1016/j.copbio.2019.08.005>). We opted not to discuss these examples in detail as they are very limited in scope (only a single metabolite) and due to the editor's request to reduce the number of references.

Reviewer #2 (Comments for the Author):

mSystems00259-20 - Emergence of metabolite provisioning as a byproduct of evolved biological functions

The opinion/hypothesis "Emergence of metabolite provisioning as a byproduct of evolved biological functions" by Gude, Pherribo, and Taga describes the concept that metabolite provisioning is not an evolutionary adaptive phenomenon but rather a natural biological consequence. The authors describe two examples of metabolic provisioning mechanisms as the main mechanisms that allowed the emergence of inter-dependencies. The manuscript is well written and easy to follow. However, the idea of leakiness by itself allowing the emergence of metabolic dependencies, seems very unlikely. The authors should point out the contradictions and disadvantages of adaptive mechanisms, contra pointing both hypotheses. Why they could not happen concomitantly?

We agree with the general assessment that the emergence of metabolic dependencies should be unlikely. We revised the text accordingly, and now clearly state that we envision the emergence of metabolic dependencies due to leakiness to be rare, yet possible, events. Our line of reasoning was inspired by analyses indicating the presence of auxotrophies in free-living microbes (<https://doi.org/10.1111/evo.12468>). Notably, free-living microbes in this dataset contain far fewer auxotrophies than endosymbionts, which is in agreement with the reviewer's assessment of 'unlikeliness'. We also address the issue of adaptive evolution versus genetic drift in these systems:

Line 53-57: 'Auxotrophy and extensive genome reduction commonly occur in obligate endosymbionts, driven by genetic drift caused by genetic isolation and the bottlenecks that occur during transmission. Auxotrophy also commonly occurs in free-living microbes, though to a lesser extent, suggesting that auxotrophy can also arise via natural selection. In these relatively rare cases, auxotrophy may be supported by leakiness of other organisms in the environment (10).'

We agree that various mechanisms, both adaptive and non-adaptive, can lead to metabolic dependencies. Depending on lifestyle and habitat, some mechanisms may be more likely than others. Generally, many of these processes may occur concurrently. We altered the text in the conclusion section to highlight these points:

Line 211-214: 'Metabolite provisioning for the benefit of others can arise through various adaptive and non-adaptive mechanisms. For example, metabolites can be provided in the context of co-evolved partnerships, where evolved reciprocity or genome reduction leads to metabolic interdependence (Fig. 1 C,D).'

Major comments:

Lines 20 and 51: The generic use of "nucleic acid auxotrophy" can be misleading. To my knowledge all organisms are autotrophic for DNA and RNA. Please review the text and provide more explanation, references.

Response: We changed "nucleic acids" to "nucleobases" throughout the manuscript. Additionally, we are citing two references with examples of auxotrophy for nucleobases (line 51).

Lines 61-62: The authors highlight the absence of knowledge of the mechanisms used by microbes to share nutrients. To date, it is known that most nutrients are shared through direct uptake from the outer environment. The authors cited a manuscript describing formation of nanotubes between microbes. It seems an odd choice since it is not known to be a common mechanism for exchanging intra-species.

Response: We agree with the reviewer and removed the citation. Furthermore, we replaced the term 'shared' with 'released' in line 62.

Line 124: "Host population" usage is confusing here. Make clear you are talking about phages bacterial hosts. In addition, what would it be, bacterial species, strains?

Response: We replaced 'host population' with 'microbial population' (line 125).

Line 124-131: How kill-the-winner hypothesis supports microbiota homeostasis? We have several examples of how stable microbial communities can be over time, i.e, the most abundant members stay the same unless perturbations happen. The idea of many uncoordinated lysis events occurring is intriguing, could the author suggest an approach to validate it?

Response: Microbial communities are indeed found to be very stable in many habitats unless perturbed. Nevertheless, we argue that viral lysis in marine environments may be viewed as a (continuously occurring) significant perturbation to a microbial system. To emphasize this, we reworded the text:

Line 92-94: 'In marine environments, viruses lyse an estimated 20-40% of microbial biomass each day, making them major drivers of phytoplankton mortality (21).'

To our knowledge direct evidence pointing to coupled oscillation of bacteria and phage are limited (see also <https://doi.org/10.3410/B4-17>). We hope that improved time-scale resolution and in situ measurements of microbial and viral abundances in natural habitats, such as the ocean, will provide useful insight into these dynamics. We changed the text and now specifically mention the limited experimental evidence of 'kill-the-winner' dynamics.

Line 132-135: 'To date, limited experimental evidence for coupled oscillations within virus-microbe systems exists; in situ measurements of microbial and viral abundances with improved time-scale resolution could provide useful insight into these dynamics (29).'

Line 138: "various viral and non-viral lysis processes", the authors should briefly cite them and explain why they are less relevant than the viral lysis process.

Response: We revised the text accordingly and now explicitly mention (and cite) examples of non-viral lysis processes:

Line 138-145: 'However, many other mechanisms of cell lysis exist and can also contribute to nutrient provisioning. For example, bacterial lysis is occasionally activated from within the host genome via prophage induction (32), and lysis is also caused by non-viral mechanisms, such as autolysis induced by stress or as part of a developmental process, and killing due to microbial competition (Fig. 3 A,C) (15-18). Autolysis has been shown to be important in supporting biofilm development by releasing DNA to form a structural component of biofilm extracellular matrices; this process could also support neighboring cells through released nutrients (19). Thus, various combinations of these processes within large and diverse microbial communities can produce a sustained nutrient supply by creating many interspersed nutrient release events.'

Lines 145, 67-68: The concept of the overproduction of metabolites has to be better explained. In regulatory processes, small changes in metabolites concentrations can dramatically affect the organism response and byproducts can impair growth (PMID: 30988035, 18618908) Additionally, how microbes would handle growth inhibition in the first place? Same question came when I was reading the two criteria for the occurrence of the initial evolutionary step (Line 73). Overproductions of metabolites and sustained supply of nutrients without a efflux route would promote growth inhibition.

Response: We revised the text and now draw parallels between the already established concept of overflow metabolism (i.e. the release of metabolites like acetate to avoid growth inhibition) and the idea of release of intracellular metabolites as a consequence of overproduction of amino acids, cofactor, and nucleobases. We hope that these changes clarify how overproduction and metabolite release are interconnected to relieve the cell of potential growth-inhibiting metabolite accumulations:

Lines 162-164: 'These processes thus perform a function akin to the release of temporarily undesirable metabolites during overflow metabolism (Fig. 2B), as both protect the cell from growth inhibition due to metabolite accumulation (7, 8).'

Additionally, we note that multiple studies provide hints that release of intracellular metabolites (or their overflow metabolism) is a common strategy of microbes to maintain metabolite, and thus growth, homeostasis. For example, Campbell et al, 2018

(<https://doi.org/10.1016/j.coisb.2017.12.001>) state: “Prominent examples for this have been shown in *Escherichia coli* that, remarkably, lacks a degradation pathway for many expensive amino acids, leading to metabolite export being the sole option to avoid accumulation beyond their optimal concentration range [36-38].” Furthermore, Reaves, 2013 (<https://doi.org/10.1038/nature12445>) established for *E. coli* that “pyrimidine homeostasis involves dual regulatory strategies, with classical feedback inhibition enhancing metabolic efficiency and directed overflow metabolism ensuring end-product homeostasis.” Due to the editor’s prompt to reduce the number of references we opted to limit our discussion of these examples in the manuscript to the following revision.

Line 176-179: ‘One strategy for relieving metabolite accumulation is degradation. However, some organisms such as *E. coli* lack degradation pathways for certain metabolites, and therefore must remove excess metabolites by exporting them (38).’

We now specify in the introduction that a metabolite release route must be available. We also draw parallels to the established phenomenon of overflow metabolism as mentioned above.

Line 73-78: ‘For this initial evolutionary step to occur in free-living microbial populations, a route for metabolite release must be present. In addition, two criteria must be fulfilled. First, the mechanism of overproducing these desirable metabolites may not incur a fitness cost to the producer, since there is no reciprocation to compensate for it. Second, a sustained supply of the metabolite must be maintained in order for auxotrophs to become fixed in the population.’

Minor comments:

Line 43: "biopolymer degradation" as a naturally occurring process outside the cell. It seems unusual for me and the references listed do not support it.

Response: We changed ‘biopolymer degradation’ to ‘polysaccharide hydrolysis’ and replaced the reference with a more appropriate one. The new reference refers to a cellulose-degrading bacterium.

Line 40-43: ‘Functions that change the extracellular environment, such as degradation of toxins, polysaccharide hydrolysis, and release of iron-scavenging siderophores, are inherently leaky, and therefore organisms that do not perform these functions can still reap their benefits (Fig. 2A) (4-6).’

Line 80: The use of the term "side effect" for byproducts is not adequate, it gives an impression of harm. Maybe replace by "final result" or "output".

Response: We changed ‘side effect’ to ‘secondary effect’ (line 81).

Line 89: No references for biofilm formation, all sporulation.

Response: We thank the reviewer for pointing this out. We included a biofilm reference (<https://doi.org/10.1128/AEM.70.6.3232-3238.2004>) in line 90.

Line 222: Check references format. No capital letter and the bacterial names should be italicized.

Response: We have corrected these formatting errors.

May 14, 2020

Dr. Michiko E Taga
University of California, Berkeley
Plant & Microbial Biology
111 Koshland Hall
Berkeley, CA 94720-3102

Re: mSystems00259-20R1 (Emergence of metabolite provisioning as a byproduct of evolved biological functions)

Dear Dr. Michiko E Taga:

Thank you for incorporating the reviewer feedback in your revised manuscript. I am pleased to let you know your manuscript has been accepted, and I am forwarding it to the ASM Journals Department for publication. For your reference, ASM Journals' address is given below.

Before it can be scheduled for publication, your manuscript will be checked by the mSystems senior production editor, Ellie Ghatineh, to make sure that all elements meet the technical requirements for publication. She will contact you if anything needs to be revised before copyediting and production can begin. Otherwise, you will be notified when your proofs are ready to be viewed.

Sincerely,

Benjamin Wolfe
Editor, mSystems

Journals Department
Phone: 1-202-942-9338